# What is the evidence for efficacy, effectiveness and safety of surgical interventions for plantar fasciopathy? A systematic review

C. Sian MacRae[1,2], Andrew J. Roche[3], Tim J. Sinnett[3], Neil E. O'Connell[1]*

**1** Department of Health Sciences, College of Health, Medicine and Life Sciences, Brunel University London, Uxbridge, United Kingdom, **2** Department of Physiotherapy, Therapy Services, Chelsea and Westminster Hospital NHS Foundation Trust, London, United Kingdom, **3** Department of Orthopaedics, Foot and Ankle Unit, Chelsea and Westminster Hospital NHS Foundation Trust, London, United Kingdom

* neil.oconnell@brunel.ac.uk

**Data Availability Statement:** All relevant data are within the manuscript, presented in Tables 1 and 2, and its Supporting Information files (Appendices S1–S7).

## Abstract

### Aims

To systematically review randomised control trials (RCT's) assessing the effectiveness and safety of surgical interventions in adults with plantar fasciopathy (PF).

### Materials and methods

We searched MEDLINE, EMBASE, Web of Science, Google Scholar, the Cochrane Central Register of Controlled Trials, trial registries and references lists. RCT's comparing surgical interventions with non-surgical or surgical comparisons in adults with PF were included. Primary outcomes were changes in first step pain severity/intensity, and incidence/nature of adverse events. Secondary outcomes included foot and ankle related disability/function, health related quality of life, cost effectiveness, changes in other reported measures of pain and medication use. Data were extracted at short-term (≤3 months post-intervention), medium-term (>3months—≤6 months post-intervention) or long-term (>6 months—≤2 years post-treatment). Certainty of evidence was evaluated using the modified Grading of Recommendations Assessment, Development and Evaluation (GRADE).

### Results

From 3620 screened records, we included 8 studies comprising 345 patients. Substantial variation across trials precluded meta-analysis, hence a narrative synthesis was conducted. We judged all studies to have high risk of bias. For all outcome comparisons our GRADE judgement for the certainty of the evidence was very low. Three studies compared one type of surgery with another largely showing little to no difference in outcomes for pain, function or quality of life. Five studies compared surgery with non-surgical interventions—three providing very low certainty evidence that surgery may improve pain and function at long-term follow-up compared to non-surgical comparisons, whereas two studies provided no long-term between-group differences. Reporting of adverse events were inadequate, inconsistent or absent across all studies.

**Funding:** The authors received no specific funding for this work.

**Competing interests:** The authors have declared that no competing interests exist.

## Conclusion

There is a paucity of high certainty evidence to support or refute the effectiveness and safety of surgical interventions in the management of PF. We make recommendations for improving the evidence base in this field.

## Introduction

Plantar fasciopathy (PF) is a degenerative condition of the plantar fascia, secondary to repetitive overloading. PF is characterized by symptoms of pain during weight-bearing activities, confined to the insertion of the plantar fascia at the antero-medial aspect of the calcaneum [1]. Diagnosis of PF is typically made through clinical examination, with common features including pain on first few steps on waking or after prolonged sitting; pain on palpation of the medial plantar aspect of the calcaneus; plantar heel pain on passive dorsiflexion of the ankle and/or toes; and pain that worsens as the day progresses [2].

PF affects approximately 10% of adults during their lifetime [3] with peak incidence of PF occurring between the ages of 45 and 64 years [4]. There is a paucity of high quality evidence to support most proposed risk factors for PF [5]. Populations at risk, supported by strong evidence, include people who are overweight or obese [4,6], or have calf tightness [7]. Risk factors, supported by weak evidence, include pes planus [8,9] or pes cavus feet [10], long-distance runners [5], and people with occupations requiring prolonged standing [3–11].

For the majority, PF is self-limiting with greater than 80% of those affected gaining complete resolution within one year [12,13]. However, persistent symptoms develop in approximately 10% of cases with detrimental effects on health related quality of life (HRQoL) [14]. Difficulty walking may affect a person's ability to maintain a healthy weight, exercise, work and has been linked to anxiety and depression [15]. Hence, determining effective treatment approaches for persistent PF is essential.

Clinical practice guidelines for first-line treatment of PF recommend conservative management [2,16]. Although multiple conservative treatment options are available, such as gel heel pads, exercise, taping and extracorporeal shock wave therapy (ECSWT) [16–18], long-term effectiveness for many is uncertain or minimal. Surgical procedures, such as plantar fasciotomy [19] or proximal medial gastrocnemius release (PMGR) [20] may be offered to people with persistent PF who's symptoms have not resolved following a 6–12 month trial of conservative management [18,21]. However, to date there are no systematic reviews of the effectiveness of these various surgical procedures for PF.

### Objectives

The primary aim of this systematic review was to determine the effectiveness and safety of surgical interventions in adults with PF.

## Methods and analysis

We registered this systematic review on Prospero (registration number: CRD42019133563) and have published its full protocol [22]. We used the following criteria for selecting studies for this review:

## Types of studies

We included RCT's published in any language. We excluded studies in which participants were not randomised to intervention groups.

## Types of participants

Studies involving adults, aged 18 years or older, diagnosed with PF, or with an alternative diagnostic label for this condition e.g. plantar fasciitis, plantar heel pain, were included. We included studies regardless of symptom duration or whether radiological diagnostic imaging had been employed.

## Types of interventions

Any surgical procedure delivered as either a stand-alone treatment compared with placebo, no treatment, usual care or another intervention, or varying surgical procedures compared with each other were included. Trials of surgery combined with another intervention were only included if the comparisons allowed for the specific evaluation of the effect of the surgery (e.g. surgery and rehabilitation versus rehabilitation only).

## Types of outcome measures

**Primary outcomes.** We analysed the following primary outcome measures where such data was available:

1. Changes in pain severity/intensity for first step pain, such as visual analogue scale (VAS), numerical rating scale (NRS). Pain intensity was presented and analysed as change on a continuous scale or in a dichotomised format as the proportion of participants in each group who attained a predetermined threshold of improvement.

2. The incidence and nature of adverse events e.g. infection, plantar fascia rupture.

**Secondary outcomes.** We analysed the following secondary outcome measures where such data was available:

1. Foot and ankle related disability/function as measured by validated clinician-report and self-report questionnaires/scales.

2. Changes in HRQoL using any validated tool.

3. Cost effectiveness

4. Changes in other reported measures of pain eg: overall pain

5. Medication use

## Timing of assessment of outcomes

Primary and secondary outcomes were classified as: (i) short term ($\leq$3 months post-intervention), (ii) medium term (>3 months—$\leq$6 months post-intervention) or (iii) long term (>6 months—$\leq$2 years post-intervention). For all outcomes, the latest outcome data within each time category was used for analysis e.g. if a study reported 3-week and 6-week pain outcomes, only the 6-week data was used.

## Search methods for identification of studies

**Electronic searches.** The following electronic databases were searched up to 25[th] February 2022 from their inception using a combination of controlled vocabulary, i.e. medical subject headings (MeSH) and free-text terms to identify published articles: Cochrane Central Register of Controlled Trials (CENTRAL) in The Cochrane Library; MEDLINE (OVID); EMBASE (OVID); Web of Science (ISI); Google scholar. There were no language restrictions. All database searches were based on this strategy but adapted to individual databases as necessary. The search strategy for MEDLINE is summarised in the online supplementary file, S1 Appendix.

**Searching other resources.** We searched clinicaltrials.gov (www.clinicaltrials.gov) and the WHO International Clinical Trials Registry Platform (http://apps.who.int/trialsearch/) for ongoing trials. In addition, reference lists of retrieved articles were checked for additional studies. We sent the list of included studies to content experts to help identify any additional relevant studies.

## Data collection and analysis

**Selection of studies.** The titles and abstracts of potential trials identified by the search strategy were independently assessed by two review authors (SM and NO'C) for eligibility. If eligibility of a study was unclear from the title and abstract, the full paper was assessed. Studies that did not match the inclusion criteria were excluded. Disagreements between review authors regarding a study's inclusion were resolved by discussion; there was no need for a third reviewer. Studies were not anonymised prior to assessment.

**Data extraction and management.** Two reviewers (NO'C and SM) independently extracted data from all included studies using a standardised and piloted data extraction form. Discrepancies and disagreements were resolved by consensus without need for a third reviewer.

We extracted the following data from each study included in the review: country of origin; study design; study population (diagnosis, diagnostic criteria used, symptom duration, age range, gender split); details of concomitant treatments that may affect outcome; sample size—active and control/comparator groups; attrition rates by group for each follow-up point; intervention(s) (including surgery type, type of surgeon, surgical approach, method of anaesthesia); rehabilitation post-surgery (including post-surgical care, rehabilitation programme received); type and details of comparator intervention (including content, delivery, duration and dose where appropriate); outcomes (primary and secondary); timepoints assessed (for the comparisons of interest to this review); industry or other financial sponsorship; author conflict of interest statements.

## Assessment of risk of bias in included studies

Two authors (SM and NO'C) independently assessed risk of bias (RoB) for each study, using the Cochrane RoB Tool [23] with any disagreements resolved by discussion. There were no cases where consensus could not be achieved. Details for how judgements were made for each domain can be found in online supplementary file, S2 Appendix.

## Measures of treatment effect

We planned to express the size of treatment effect on pain intensity, as measured with a VAS or NRS, using the mean difference (where all studies utilised the same measurement scale) or the standardised mean difference (where studies used different scales). However, it was not possible to pool any of the studies in this review.

We had planned to calculate Risk Ratio and Risk Difference with 95% confidence intervals for dichotomised outcome measures in addition to the number needed to treat to benefit and harm as an absolute measure of treatment effect, however, this was not possible.

## Unit of analysis issues

Where an included trial compared multiple treatment arms to the same control and those arms were included in the same meta-analysis, we planned to split the number of participants in the control treatment arm between those treatment arms. However, based on the studies in this review, this was not required.

## Dealing with missing data

Where insufficient data were presented in the study report, we contacted study authors to request access to the missing data.

## Assessment of heterogeneity

We planned to not combine studies that compared surgery to no treatment/ usual care with studies that compared surgery to sham/ placebo in the same analysis, to assess heterogeneity using the $Chi^2$ test to investigate the statistical significance of such heterogeneity, and the $l^2$ statistic to estimate the amount of heterogeneity. However, based on the studies within the review, assessment of heterogeneity was not indicated.

## Assessment of reporting biases

We planned to use funnel plots to visually explore the likelihood of reporting biases when there were at least 10 studies in a meta-analysis and included studies differed in size, and planned to use Egger's test to detect possible small study bias. However, as no meta-analysis was conducted this was not indicated.

## Data synthesis

We intended to pool results where adequate data existed using Review Manager (RevMan 5.3, 2014). We planned to perform separate meta-analysis for the following classes of surgery: plantar fasciotomy, PMGR at the following time points: short-term ($\leq$ 3 months post-intervention), medium-term ($>$3months—$\leq$6 months post-intervention) or long-term ($>$6 months —$\leq$2 years post-treatment). For each broad class of surgery we planned to conduct the following comparisons where adequate data were available: surgery versus sham surgery, surgery versus minimal care/ waiting list/ no treatment, surgery versus non-surgical treatment. For all analyses, we presented explicitly and clearly, the outcome of the RoB assessments for included studies in the reporting. Insufficient data were found to support statistical pooling, hence we conducted narrative synthesis of the evidence.

## Certainty of the evidence

We assessed the overall certainty of the evidence for each outcome using the GRADE approach [23]. Two review authors (SM and NOC) independently rated the quality of the evidence for each planned comparison. Further information on factors that may decrease the certainty of the evidence and the GRADE system are provided in the online supplementary file, S3 Appendix.

## Subgroup analysis and investigation of heterogeneity

Where substantial heterogeneity was found ($I^2 > 50\%$, $p < 0.10$) we planned to conduct subgroup analysis investigating the possible impact of the type of surgical intervention (e.g. fasciotomy versus PMGR) or surgical approach (open vs endoscopic). However, based on the studies within the review, this analysis was not possible.

## Sensitivity analysis

Where sufficient data was available we planned to conduct sensitivity analyses on RoB, investigating the effect of including/ excluding studies rated at high RoB (on one or more criteria other than blinding of patients or care providers) from the analysis and the choice of meta-analysis model (investigating the impact of applying a fixed-effects instead of a random-effects model). However, based on the studies within the review, this was not conducted.

## Results

### Description of studies

See Table 1 for the *'Characteristics of included studies'*.

### Results of the search

We searched the literature up to 25[th] February 2022. The search identified 3620 articles of which 2349 unique articles were identified after duplicates were removed. A total of 2326 articles were excluded after screening title and abstract, an additional 13 articles were excluded after full-text screening. We requested full study reports from the authors of one eligible published abstract [33] and two registered clinical trials but did not receive a response. Finally, nine published RCT's from eight unique studies met the inclusion criteria and were included in this review [24–32]. Fig 1 presents a flow diagram outlining the trial screening and selection process.

### Excluded studies

Information relating to the twelve trial reports that were excluded can be found in the online supplementary file, S4 Appendix. The reasons for exclusion included that the trials were either not RCT's (n = 6), not investigating a surgical intervention (n = 4), were withdrawn trial protocols (n = 1) and that full text was not available (n = 1).

### Included studies

Details of all included studies are presented in Table 1. We contacted four authors of included studies for additional data that could not be extracted from the papers. Two authors responded providing the additional data requested [31,32]. One author confirmed that two papers [29,30] contained data from the same study population, the more recent publication providing a longer follow-up timepoint [30]. One author did not respond following a request for additional data [28].

We found 3 registered but unpublished clinical trials (anticipated combined n = 117). These compared: open plantar fasciotomy versus conservative treatment, consisting of stretching and strengthening exercises, in people with plantar fasciitis of greater than 6 months duration, reported as 'not yet recruiting' (National clinical Trial number: NCT05066919); endoscopic isolated gastrocnemius recession versus combined endoscopic gastro-soleus recession in people with isolated gastrocnemius contracture, reported as 'ongoing' (Thai Clinical

**Table 1. Characteristics of included studies.**

| Author | Setting | Inclusion criteria | No. of participants | Female (%) | Interventions | Age in years mean (SD and/or *range*) | Duration of complaint in months mean (SD and/or *range*) | Outcomes | Follow-up timepoints measured |
|---|---|---|---|---|---|---|---|---|---|
| Catal et al. [24] | Unclear | Single site heel pain with local pressure at the origin of the proximal plantar fascia, failure of 3 lines of conservative treatment during previous 6 months. | 41 | 70.7 | Deep fascial approach fasciotomy | 52.43 (6.98) | 21.10 (6.81) | AOFAS-AHS/ 100 | 3 weeks 3 months 6 months 12 months |
| | | | | | Superficial fascial approach fasciotomy | 51.3 (7.91) | 20.05 (8.78) | VAS pain/10 | 3 weeks 3 months 6 months 12 months |
| | | | | | | | | Early complications | "Immediately after the procedure" "During the follow-up period" |
| | | | | | | | | Late complications | |
| Molund et al. [25] | Unclear | Aged 18–70 years, plantar heel pain >12 months, treatment. Diagnosis based on clinical symptoms: pain at first step in the morning, pain on palpation of the plantar fascia insertion on the calcaneus. Isolated contracture of the gastrocnemius (+ve Silfverskiöld test) | 40 | 77.5 | Proximal Medial Gastrocnemius Recession and stretching | 46 (*29–68*) | 31 (*12–252*) | AOFAS/100 | 3 months 12 months |
| | | | | | Stretching | 45 (*22–63*) | 33(*12–396*) | VAS Pain/10 | 3 months 12 months |
| | | | | | | | | SF-36 | 12 months |
| | | | | | | | | Complications | Not clearly described |
| Othman et al. [26] | Unclear | Unilateral symptoms, ≥ 6 months duration, no response to conservative treatment | 50 | Not reported | Endoscopic plantar fasciotomy | 39.14 (*22–51*) | 10.96 (*6–23*) | AFOAS/100 | 6 weeks 12 weeks 24 weeks Then 3 monthly intervals till end of study |
| | | | | | Platelet rich plasma injection | 36.04 (*25–49*) | 11.59 (*6–34*) | VAS pain/10 | 6 weeks 12 weeks 24 weeks Then 3 monthly intervals till end of study |
| | | | | | | | | Complications | Unclear |
| Radwan et al. [27] | Unclear | Single site heel pain at origin of proximal plantar fascia, failure of at least 3 conservative treatment measures in last 6 months. VAS for pain > 40/100 after first 5 minutes of walking in the morning | 65 | 38.5 | Endoscopic plantar fasciotomy | 37.7 (9.42) | 18 (10.9) | VAS Morning pain: /100 | 3 weeks 12 weeks 12 months |
| | | | | | Extracorporeal shockwave therapy | 39.7 (8.79) | 17.45 (8.5) | AOFAS/100 | 3 weeks 12 weeks 12 months |

(*Continued*)

**Table 1.** (Continued)

| Author | Setting | Inclusion criteria | No. of participants | Female (%) | Interventions | Age in years mean (SD and/or *range*) | Duration of complaint in months mean (SD and/or *range*) | Outcomes | Follow-up timepoints measured |
|---|---|---|---|---|---|---|---|---|---|
| Sadak et al. [28] | Unclear | Resistant plantar fasciitis after ≥6 months conservative management, diagnosis confirmed radiologically (presence of calcaneal spur, perifascial oedema, increased plantar fascial thickness of >4 mm). Age 30–60. | 33 | 90.9 | Lateral plantar nerve release with drilling | 43.9 (8.4) | 17.1 (6.9) ("time to surgery") | Modified Mayo scoring system for plantar fasciotomy<br><br>Complications | Mean follow-up 27 months (no specific time points reported)<br><br>Unclear |
| | | | | | Lateral plantar nerve release without drilling | 46.3 (7.5) | 18.7 (7.5) ("time to surgery") | | |
| Gamba et al. [29,30] | Unclear | >18 years of age, clinical diagnosis of PF of ≥ 9/12, confirmed by MRI/US; non-responsive to conservative management (anti-inflammatory drugs, stretching calf and fascia, physiotherapy, insoles, LA and steroid injection, +/- ECSWT.) | 38 | 78.51 | Open plantar fasciotomy | 51.3 (11.4) | 32.1 | VAS Pain<br><br>SF-36<br><br>AOFAS-AHS | 1 month 3 months 6 months 12 months<br><br>1 month 3 months 6 months 12 months<br><br>1 month 3 months 6 months 12 months |
| | | | | | Proximal medial gastrocnemius release | 46.2 (11.1) | 27 | | |
| Catal et al. [32] | Unclear, likely secondary care | Diagnosis of Plantar fasciitis, treated conservatively for ≥ 6 months with no response to ≥ 3 conservative treatment modalities. | 43 | 55.81 | Endoscopic Plantar fascia release | 53.5 (6.7, *39–67*) | 22.3(7, *10–34*) | AOFAS-AHS<br><br>Early complications<br><br>Delayed complications | 3 weeks 3 months 6 months 12 months<br><br>"Immediately after procedure"<br><br>"Developed during follow-up period" |
| | | | | | Cryosurgery | 46.9 (8.9, *32–60*) | 20.7 (15, *6–48*) | | |
| Johannsen et al. [31] | Private rheumatology clinic and university clinic | Clinical PF symptoms ≥ 3/12 (1st step pain; tenderness on palpation medial calcaneal attachment, PF ≥ 4mm thick on US). Aged 20–65. Danish speaking. | 30 | 67.86 | Endoscopic partial plantar fasciotomy | 49 (8) | 25 (*4–180*) | VAS First step pain<br><br>FFI<br><br>VAS pain during activity | 3 months 6 months 12 months 24 months<br><br>6 months 12 months 24 months<br><br>3 months 6 months 12 months 24 months |
| | | | | | Corticosteroid injection | 45 (5) | 15 (*4–36*) | | |

Legend: VAS: Visual analogue scale; AOFAS-AHS: American Orthopaedic Foot and Ankle Society Ankle hind-foot score; FFI: Foot functional index; SF-36:Short Form 36. US: Ultrasound; MRI: Magnetic resonance imaging.

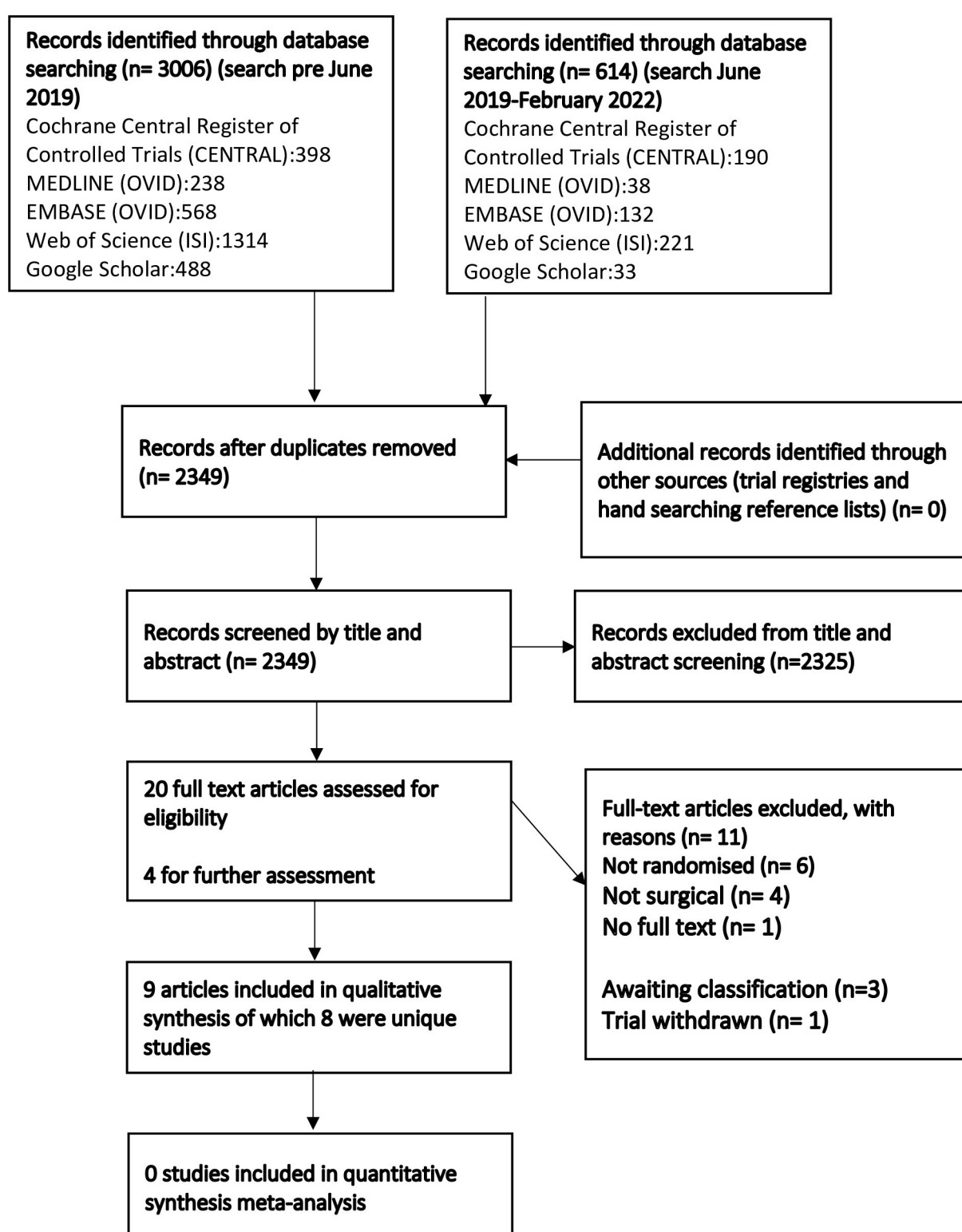

**Fig 1. Flow diagram outlining the trial screening and selection process.**

Trials Registry Number: 20180212001); and coblation via percutaneous fasciotomy versus standard surgical fasciotomy, reported as 'completed' (National Clinical Trial number: NCT00189592). We contacted authors of the latter two trials for data, however, received no response. We classified these trials as 'awaiting classification' (Online supplementary file, S4 Appendix).

## Characteristics of the included studies

**Design.**    All 8 studies were RCT's using a parallel group design and two intervention arms.

**Interventions and comparisons.**    The interventions delivered in each trial are presented in Table 1. No studies compared a surgical procedure to a sham procedure, and none compared surgery to a minimal care/waiting list/no treatment group. Three studies compared one type of surgery with another: endoscopic deep fascial approach fasciotomy (DFAF) compared to endoscopic superficial fascial approach fasciotomy (SFAF) [24]; lateral plantar nerve release with drilling compared to without drilling [28]; open plantar fasciotomy compared to proximal medial gastrocnemius release (PMGR) [29,30]. Five studies compared surgery with a non-surgical intervention: PMGR and stretching compared to stretching alone [25]; endoscopic plantar fasciotomy (EPF) compared to platelet rich plasma (PRP) injection [26]; EPF compared to ECSWT [27]; EPF with heel spur removal compared to corticosteroid injections [31]; and EPF compared to cryosurgery [32]. For further detail on study characteristics, see online supplementary file, S5 Appendix.

## Outcomes considered across studies

**Primary outcomes.**    *First step pain.* Two studies reported outcomes for first step, or morning pain [27,31]. Both authors reported data at 3- and 12-months post-intervention. In addition, Johannsen et al. [31] reported data at 6-months and 24-months post-intervention.

*Adverse events.* Six studies reported information regarding adverse events [24–26,28–30,32]. The level of detail reported varied considerably between studies. Two studies [24,32] described how early and late adverse events were classified. Four studies did not report methods used for measuring and classifying the adverse events reported [25,26,29,30]. Two studies did not provide information regarding adverse events [27,31].

**Secondary outcomes.**    *Foot and ankle-related disability or function.* All included studies assessed disability or function. Six studies used the American Orthopaedic Foot and Ankle Society Ankle Hindfoot Scale (AOFAS-AHS) [24–27,29,30,32], one study used the Foot Functional index (FFI) [31], and one the Modified Mayo scoring system for PF [28].

*Health related quality of life.* Two studies reported measures of HRQoL (Short Form (36) Health Survey Questionnaire [SF-36]) [25,29,30]. Six studies did not report HRQoL outcomes [24,26–28,31,32].

*Cost-effectiveness.* None of the studies included within this review reported cost-effectiveness.

*Other reported measures of pain.* Five studies reported VAS pain outcomes with descriptors other than first step/morning pain. These include 'worst pain in 24 hours' [25], 'pain during activity' [31] and 'VAS for pain' with no further descriptor provided [24,26,29,30]. Two studies did not report pain outcomes [28,32].

*Medication use.* One study reported on short term (15 days) medication use [29,30].

None of the included studies reported responder data or analyses in terms of the proportion of participants who met a predetermined threshold for symptom improvement or a predetermined endpoint.

### Risk of bias in the included studies

A summary of the RoB assessments for included studies is presented in Figs 2 and 3 (composed in Review Manager (RevMan) Version 5.4., The Cochrane Collaboration, 2020). We considered all eight studies to have a high RoB [24–32] with all studies judged at high or unclear RoB on more than one domain. For more detail on RoB judgements across the domains see online supplementary file, S6 Appendix.

### Certainty of the evidence

All comparisons were based on single small trials at high RoB. For all comparisons our GRADE judgement for the certainty of the evidence was very low, downgraded twice for limitations of studies, and once for imprecision and inconsistency.

### Effects of interventions

A summary of all results for each comparison is given in Table 2. We used Review Manager (Revman, Version 5.4, The Cochrane Collaboration, 2020) for estimating effect sizes. For more detailed information regarding secondary outcomes, see online supplementary file, S7 Appendix.

**Endoscopic deep fascial approach fasciotomy versus endoscopic superficial fascial approach fasciotomy.** One study [24] (n = 41) with a high RoB compared DFAF versus SFAF.

*Primary outcomes. Adverse events*: Two participants reported early complications (haematoma) and two reported late complications in the DFAF group (recalcitrant heel pain, positive Tinel's sign). There were no reported complications in the SFAF group.

*Secondary outcomes.* Both groups demonstrated improvements in function (AOFAS) at short-, mid- and long-term follow-up. There were no clear between-group differences at any follow-up timepoint. Pain scores were lower in the SFAF group compared to the DFAF group at short-term follow-up; there was no clear between-group differences in pain at mid- or long-term follow-up.

The authors concluded there were no long-term benefits of one type of surgery compared to the other for pain or function but that SFA appeared to be safer due to the lower complication rate.

**Proximal medial gastrocnemius recession and stretching versus stretching.** Molund et al. [25] (n = 40, high RoB) compared PMGR and stretching, to stretching alone in people with PF.

*Primary outcomes. Adverse events.* 'Minor' complications were described for four patients in the surgical group (three participants reported prolonged pain or swelling, one reported calf cramps). The authors did not comment on whether or not complications occurred within the non-surgical stretching group.

*Secondary outcomes.* The authors reported a between-group difference at short- (AOFAS-AHS and pain) and long-term follow-up (AOFAS-AHS, pain, and all domains of SF36) in favour of the surgical group. We were unable to calculate effect sizes/confidence intervals (CIs) as authors reported medians and range only for these comparisons.

The authors concluded that people with recalcitrant PF who undergo PMGR with a stretching programme have reduced pain and improved function at one year follow-up when compared to people managed non-surgically.

**Endoscopic plantar fasciotomy versus platelet rich plasma injection.** One study [26] (n = 50, high RoB), compared EPF with a PRP injection in people with PF. Outcomes assessed included VAS for pain, AOFAS and complications. The authors, however did not present

**Fig 2. Risk of bias summary: Review authors' judgements about each risk of bias item for each included study.**
Composed in Review Manager (RevMan) [Computer program]. Version 5.4., The Cochrane Collaboration, 2020.

effect sizes/ CIs for these comparisons. Furthermore, we weren't able to include results from this study in our analyses at predefined timepoints as even though the methods stipulated timepoints that met our inclusion criteria, their results reported one amalgamated result of assessment timepoints for pain and function (ranging from 6–42 months); hence, we were not able to extract an effect estimate for any time points of interest.

*Primary outcomes. Adverse events.* Two minor, fully resolving complications were reported in the surgical group (persistent drainage from the wound, numbness and paraesthesia in the lateral plantar nerve distribution). No complications, other than post-injection pain, were reported in the injection group.

*Secondary outcomes.* Both groups demonstrated improvements in function and pain 'over time'; there were no between-group differences 'over-time' for either outcome.

The authors concluded that both management methods gave comparable results at late follow-up.

**Endoscopic plantar fasciotomy versus extracorporeal shockwave therapy.** Radwan et al. [27] (n = 65, high RoB) compared the effectiveness of EPF with ECSWT in people with PF. Outcomes assessed included VAS for morning pain and AOFAS-AHS. The authors did not present effect sizes/ CIs for those comparisons.

*Primary Outcomes. Pain (VAS for morning pain).* Both groups demonstrated improvements at short-term (3 months) and long-term (12 months) follow-up; there were no between-group differences at either timepoint. In the EPF group at 3 months the median pain score (VAS morning pain/100) was 30 (interquartile range [IQR] 25–40) compared to 30 (IQR 20–40.75) in the ECSWT group. At 12 months the median pain score in the EPF group was 16 (IQR 11–25) compared to 15 (IQR 5–25) in the ECSWT group.

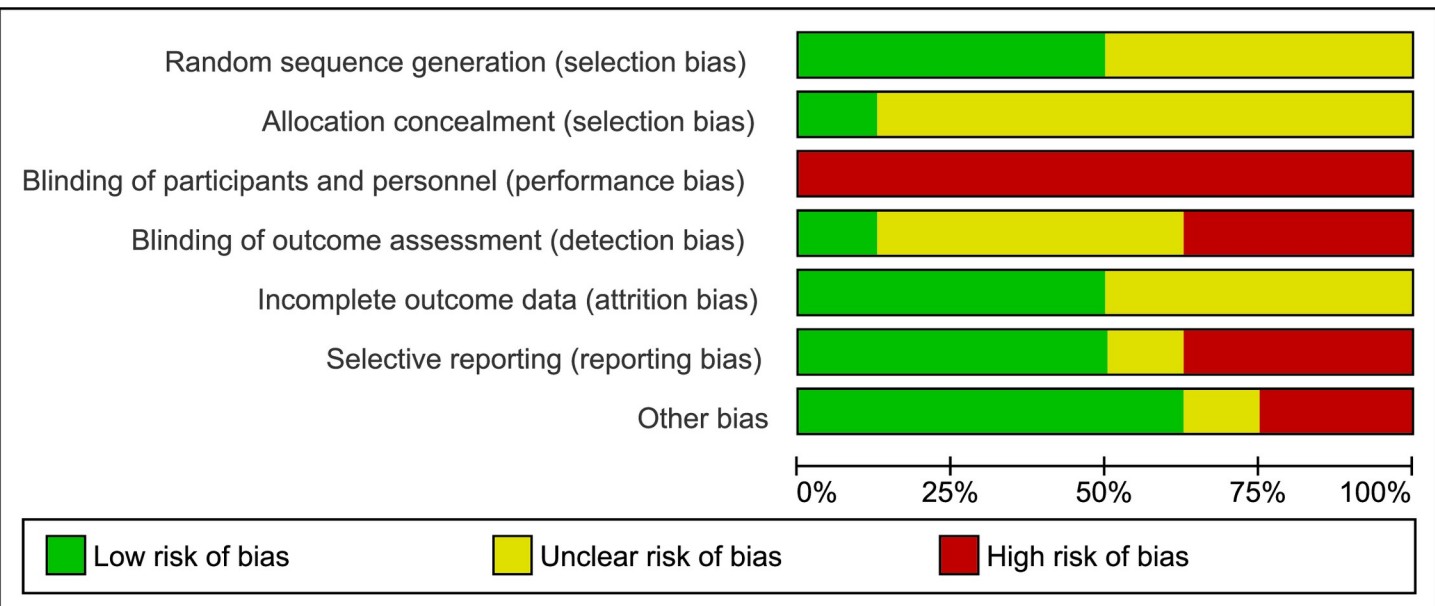

**Fig 3. Risk of bias graph: Review authors' judgements about each risk of bias item presented as percentages across all included studies.**

**Table 2. Results of included studies.**

| Comparison | Author, year of publication | Outcomes | Follow-up | Group | Number of participants (n) | Outcome Mean/median (SD/range/*25-75 percentile*) or *n(%)* |
|---|---|---|---|---|---|---|
| Deep fascial fasciotomy versus Superficial fascial fasciotomy | Catal et al (2017) [24] | AOFAS-AHS/100 | Short term (3 months) | Deep fascial fasciotomy | 21 | 73.71 (11.5) |
| | | | | Superficial fascial fasciotomy | 20 | 77.65 (9.34) |
| | | | Medium term (6 months) | Deep fascial fasciotomy | 21 | 79.9 (11.05) |
| | | | | Superficial fascial fasciotomy | 20 | 82.45 (9.26) |
| | | | Long term (12 months) | Deep fascial fasciotomy | 21 | 82.71 (10.9) |
| | | | | Superficial fascial fasciotomy | 20 | 84.7 (7.72) |
| | | VAS pain/10 | Short term (3 months) | Deep fascial fasciotomy | 21 | 2.9 (1.48) |
| | | | | Superficial fascial fasciotomy | 20 | 2.25 (1.73) |
| | | | Medium term (6 months) | Deep fascial fasciotomy | 21 | 2.14 (1.55) |
| | | | | Superficial fascial fasciotomy | 20 | 1.8 (1.63) |
| | | | Long term (12 months) | Deep fascial fasciotomy | 21 | 1.81 (1.47) |
| | | | | Superficial fascial fasciotomy | 20 | 1.5 (0.94) |
| | | Early complications | Immediately after the procedure | Deep fascial fasciotomy | 21 | 2 (9.52%) |
| | | | | Superficial fascial fasciotomy | 20 | 0 (0) |
| | | Late complications | During the follow-up period | Deep fascial fasciotomy | 21 | *2 (9.52%)* |
| | | | | Superficial fascial fasciotomy | 20 | *0 (0%)* |
| PMGR and Stretching versus Stretching | Molund et al (2018) [25] | AOFAS-AHS/100 | Short term (3 months) | PMGR + stretching | 20 | **85.5 (64–100)** |
| | | | | Stretching | 20 | **66.5 (36–85)** |
| | | | Long term (12 months) | PMGR + stretching | 20 | **88 (50–100)** |
| | | | | Stretching | 20 | **65.5 (31–88)** |
| | | VAS Worst pain in 24 hrs /10 | Short term (3 months) | PMGR + stretching | 20 | **3.3 (0–8.1)** |
| | | | | Stretching | 20 | **6.9 (2.1–10)** |
| | | | Long term (12 months) | PMGR + stretching | 20 | **2.8 (0–8.1)** |
| | | | | Stretching | 20 | **7.4 (0.2–9.3)** |
| | | Complications | Time scale unclear | PMGR + stretching | 20 | *4 (20%)* |
| | | | | Stretching | 20 | Not reported |
| | | SF-36 | Long term (12 months) | PMGR + stretching | 20 | Physical functioning: **90 (55–100)** <br> Physical role functioning: **100 (0–100)** <br> Bodily pain: **52 (20–100)** <br> General Health Perceptions: 77 **(20–100)** <br> Vitality: **68 (5–95)** <br> Social role functioning: **100 (25–100)** <br> Emotional role functioning: **100 (0–100)** <br> Mental Health: **84 (44–100)** |
| | | | Long term (12 months) | Stretching | 20 | Physical functioning: **63 (15–100)** <br> Physical role functioning: **0 (0–100)** <br> Bodily pain:**32 (0–100)** <br> General Health Perceptions: **56 (15–100)** <br> Vitality: **50 (5–100)** <br> Social role functioning: **75 (0–100)** <br> Emotional role functioning: **100 (0–100)** <br> Mental Health: **70 (24–100)** |
| Endoscopic fasciotomy versus PRP injection | Othman et al (2015) [26] | AOFAS-AHS/100 | *Mean 18.25 (range 6–42) months | Endoscopic fasciotomy | Unclear | 94 **(78–97)** |
| | | | *Mean 17.45 (range 6–40) months | PRP injection | Unclear | 92 **(78–95)** |
| | | VAS Pain/10 | *Mean 18.25 (range 6–42) months | Endoscopic fasciotomy | Unclear | 2.35 **(1–4)** |
| | | | *Mean 17.45 months (range 6–40 months) | PRP injection | Unclear | 2.9 **(1–4)** |
| | | Complications | *Mean 18.25 months (range 6–42 months) | Endoscopic fasciotomy | Unclear | 2 |
| | | | *Mean 17.45 months (range 6–40 months) | PRP injection | Unclear | 'a few cases' of post injection pain |

(*Continued*)

**Table 2.** (Continued)

| Comparison | Author, year of publication | Outcomes | Follow-up | Group | Number of participants (n) | Outcome Mean/median (SD/range/25-75 percentile) or n(%) |
|---|---|---|---|---|---|---|
| Endoscopic fasciotomy versus ECSWT | Radwan et al (2012) [27] | VAS Morning Pain/ 100 | Short term (3 months) | Endoscopic fasciotomy | 31 | **30 (25–40)** |
| | | | | ECSWT | 34 | **30 (20–40.75)** |
| | | | Long term (12 months) | Endoscopic fasciotomy | 31 | **16 (11–25)** |
| | | | | ECSWT | 34 | **15 (5–25)** |
| | | AOFAS-AHS/100 | Short term (3 months) | Endoscopic fasciotomy | 31 | **77 (72–84)** |
| | | | | ECSWT | 34 | **80.5 (73–85)** |
| | | | Long term (12 months) | Endoscopic fasciotomy | 31 | **86 (76–89)** |
| | | | | ECSWT | 34 | **87 (76.75–97)** |
| Lateral plantar nerve release with drilling versus Lateral plantar nerve release without drilling | Sadak et al (2015) [28] | Modified Mayo scoring system for plantar fasciotomy | *Mean 27.6 (SD 20.4) and 25.6 (SD 14) months; unclear which follow-up timescale represents which group). | Lateral plantar nerve release with drilling | 18 | 93.9 (6.97) *measures of variance not described* Grading: Excellent:15, Good:2, Fair:1 |
| | | | | Lateral plantar nerve release without drilling | 15 | 83 (8.2) *measures of variance not described* Grading: Excellent:6, Good:4, Fair:5 |
| | | Complications | *Mean 27.6 (SD 20.4) and 25.6 (SD 14) months; unclear which follow-up timescale represents which group. | Lateral plantar nerve release with drilling | 18 | *3 (16.7%)* |
| | | | | Lateral plantar nerve release without drilling | 15 | *1 (6.6%)* |
| Endoscopic plantar fascia release versus Cryosurgery | Catal et al. (2020) [32] | AOFAS-AHS/100 | Short Term (3 months) | Endoscopic plantar fascia release | 23 | 72.1 (10) |
| | | | | Cryosurgery | 20 | 69 (11.9) |
| | | | Medium term (6 months) | Endoscopic plantar fascia release | 23 | 82.4 (8.6) |
| | | | | Cryosurgery | 20 | 71.7 (14) |
| | | | Long term (12 months) | Endoscopic plantar fascia release | 23 | 84.6 (7.8) |
| | | | | Cryosurgery | 20 | 73.2 (15.6) |
| | | Complications | Early (immediately after procedure) | Endoscopic plantar fascia release | 23 | 0 (0) |
| | | | | Cryosurgery | 20 | 0 (0) |
| | | | Delayed (during the follow-up period) | Endoscopic plantar fascia release | 23 | 0 (0) |
| | | | | Cryosurgery | 20 | 0 (0) |
| Proximal medial gastrocnemius release (PMGR) versus Open plantar fasciotomy | Gamba et al. (2019, 2020) [29,30] | VAS for pain/100 | Short Term (3 months) | PMGR | 15 | 44.9 (31.7) |
| | | | | Open plantar fasciotomy | 21 | 27.33 (21.5) |
| | | | Medium term (6 months) | PMGR | 15 | 25.4 (19.1) |
| | | | | Open plantar fasciotomy | 21 | 33.3 (26.2) |
| | | | Long term (12 months) | PMGR | 15 | 18.3 (15.1) |
| | | | | Open plantar fasciotomy | 21 | 28.7 (25.6) |
| | | AOFAS-AHS/ 100 | Short Term (3 months) | PMGR | 15 | 87.4 (9.3) |
| | | | | Open plantar fasciotomy | 21 | 83.7 (12.5) |
| | | | Medium term (6 months) | PMGR | 15 | 89.9 (9.3) |
| | | | | Open plantar fasciotomy | 21 | 82.3 (15.9) |
| | | | Long term (12 months) | PMGR | 15 | 89 (9.9) |
| | | | | Open plantar fasciotomy | 21 | 86.7 (12.1) |
| | | SF-36 | Long term (12 months) | PMGR | 15 | Physical functioning: 43.8 (12.7) Physical role functioning: 45.6 (8.1) Bodily pain: 41.3(12.9) General Health Perceptions: 48.6 (9.0) Vitality: 51.3 (12.0) Social role functioning: 44.6 (10.2) Emotional role functioning: 47.4 (11.6) Mental Health: 46.9 (13.1) |
| | | | | Open plantar fasciotomy | 21 | Physical functioning: 46.4 (10.5) Physical role functioning: 43 (11.4) Bodily pain: 44.1 (10.7) General Health Perceptions: 46.6 (8.6) Vitality: 51.1 (13) Social role functioning: 48 (11.5) Emotional role functioning: 48.3 (11.8) Mental Health: 46 (10.7) |

(Continued)

**Table 2.** (Continued)

| Comparison | Author, year of publication | Outcomes | Follow-up | Group | Number of participants (n) | Outcome Mean/median (SD/range/*25-75 percentile*) or *n(%)* |
|---|---|---|---|---|---|---|
| | | Complications | Unclear | PMGR | 15 | 2 |
| | | | | Open plantar fasciotomy | 21 | 1 |
| Endoscopic partial fasciotomy versus Corticosteroid injection | Johannsen et al. (2020) [31] | FFI/ 230 | Short Term (3 months) | Endoscopic partial fasciotomy | 14 | 67.43 (44.62) |
| | | | | Corticosteroid injection | 14 | 48.29 (30) |
| | | | Medium term (6 months) | Endoscopic partial fasciotomy | 14 | 24.71 (18.31) |
| | | | | Corticosteroid injection | 13 | 31.85 (22.95) |
| | | | Long term (24 months) | Endoscopic partial fasciotomy | 13 | 4.23 (5.80) |
| | | | | Corticosteroid injection | 14 | 8.90 (14.51) |
| | | VAS morning pain/ 100 | Short Term (3 months) | Endoscopic partial fasciotomy | 14 | 36.07 (33.63) |
| | | | | Corticosteroid injection | 14 | 34.93 (28.28) |
| | | | Medium term (6 months) | Endoscopic partial fasciotomy | 14 | 10.43 (9.61) |
| | | | | Corticosteroid injection | 13 | 18.23 (18.75) |
| | | | Long Term (24 months) | Endoscopic partial fasciotomy | 13 | 0.00 (0.00) |
| | | | | Corticosteroid injection | 14 | 15.00 (20.94) |
| | | VAS during activity/ 100 | Short Term (3 months) | Endoscopic partial fasciotomy | 14 | 31.14 (27.69) |
| | | | | Corticosteroid injection | 14 | 28 (21.99) |
| | | | Medium term (6 months) | Endoscopic partial fasciotomy | 14 | 11.21 (12.15) |
| | | | | Corticosteroid injection | 13 | 23.85 (18.27) |
| | | | Long Term (24 months) | Endoscopic partial fasciotomy | 13 | 0.77 (2.77) |
| | | | | Corticosteroid injection | 14 | 12.50 (14.51) |
| | | Complications/side effects | No time point indicated. | Endoscopic partial fasciotomy | 14 | *0 (0%)* |
| | | | | Corticosteroid injection | 14 | *0 (0%)* |

Legend: VAS: Visual analogue scale; AOFAS-AHS: American Orthopaedic Foot and Ankle Society Ankle hind-foot score; FFI: Foot functional index; SF-36:Short Form 36.

*Results reported one amalgamated result of assessment timepoints, hence unable to extract effect estimate for any time points of interest for these studies.

*Secondary outcomes.* Both groups demonstrated improvements in function at short-term and long-term follow-up; there were no between-group differences at either timepoint.

**Lateral plantar nerve release with drilling versus lateral plantar nerve release without drilling.** One study [28] (high RoB, n = 33) compared lateral plantar nerve release with and without calcaneal drilling for people with persistent PF. Outcomes assessed post-intervention were complications and function (modified Mayo scoring system for PF). We were not able to include results from this study in our analyses at predefined timepoints as they reported an amalgamated result of assessment timepoints for the two groups, hence, we were not able to extract an effect estimate for any timepoint of interest.

*Primary outcomes. Adverse Events.* Three adverse events were reported in the 'with drilling' group and one adverse event in the group 'without drilling'. Although complications were presented (heel numbness, foot oedema, two cases of superficial wound infection) it was not possible to interpret which group each complication related to.

*Secondary outcomes.* The authors reported the group receiving drilling had a better outcome for function at follow-up than those not receiving drilling.

The authors conclude that the addition of calcaneal drilling to release of the lateral plantar nerve results in better outcome than release alone.

**Open plantar fasciotomy versus proximal medial gastrocnemius release.** Gamba et al, [29,30] (high RoB, n = 38) compared the effectiveness of open plantar fasciotomy with PMGR in people with recalcitrant PF. Outcomes assessed included VAS for pain, SF-36, AOFA-S-AHS, medication use and complications.

*Primary outcomes. Adverse events.* Two minor, fully resolving complications were reported in the PMGR group (superficial wound infection, sural nerve lesion); one minor, fully resolving complication (dehiscence of the wound) was reported in the fasciotomy group.

*Secondary outcomes.* Although both groups demonstrated improvements in pain, HRQoL, and function at long-term follow-up, no between-group differences were noted for any of the outcomes assessed. Although no data were presented, the authors report that, in the first 15 days post-surgery, there was no between-group difference in opioid (tramadol) consumption.

The authors conclude that both surgical approaches are safe and effective, recommending PMGR as the technique of choice due to a faster recovery period and its potential for less biomechanical consequences.

**Endoscopic partial plantar fasciotomy with heel spur removal versus controlled non-operative treatment (corticosteroid injections).** Johannsen et al. [31] (n = 30, high RoB) compared EPF and strength training with a series of three corticosteroid injections and strength training in people with PF. Outcomes assessed included the FFI and VAS for both first step pain and pain during activity. In addition, complications were reported in the results, but no clear method for evaluating adverse events was presented.

*Primary outcomes. Pain–morning pain.* No between-group differences were noted in the short- (effect size [95% CI]: 1.14 [-21.86, 24.14]), and mid-term (effect size [95% CI]: -7.80 [-18.84, 3.24]) follow-up 'first step pain' VAS scores. A lower (improved) pain score in the fasciotomy group was noted at 24 months (effect size [95% CI]: -15.00 [-26.38, -3.62]).

*Adverse events.* No 'severe' adverse events were reported in either group.

*Secondary outcomes.* No between-group differences were noted in the short-, mid-, or long-term follow-up FFI scores, or in the short-term for 'pain during activity'. A lower (improved) VAS score was noted in the fasciotomy group at mid- and long-term follow-up.

The authors concluded that although both groups improved over time, EPF with spur removal appeared superior to corticosteroid injections at long-term follow-up.

**Endoscopic plantar fascia release versus cryosurgery.** Catal et al. [32] (n = 43, high RoB) compared EPF with cryosurgery in patients with PF that had not responded to conservative treatment over a 6-month duration. Outcome measures included AOFAS-AHS, and early and late complications.

*Primary outcomes. Adverse events.* No complications were reported in the fasciotomy group. Two participants in the cryosurgery group reported persistent heel pain at 12-month follow-up.

*Secondary outcomes.* For both groups, function improved significantly at 12 months compared to baseline, with a greater improvement at 6- and 12-months for the EPF group. No between-group differences in function were noted in the short-term.

## Discussion

### Summary of main findings

The objective of this review was to assess the effectiveness of surgery in the management of PF by assessing its influence on pain, function, HRQoL, cost effectiveness, and evaluating incidence of adverse events following the procedure. The evidence presented in this systematic review was of very low certainty and hence the true effects may be considerably different to those observed in the included studies. Our 'very low' GRADE judgements related mainly to imprecision and limitations of included studies.

In general, studies that compared different surgical interventions largely showed little to no difference in outcomes for pain, function [24,29,30] and HRQoL [29,30]. In contrast, one study, at high RoB, comparing two surgical interventions, reported very low certainty evidence

that the addition of calcaneal drilling to release of the lateral plantar nerve resulted in better outcome than release alone [28]. However, due to their results reporting one amalgamated result of assessment timepoints, we were not able to extract an effect estimate for any time points of interest in this study. The majority of studies that compared operative to non-operative interventions provided very low certainty evidence that surgery may improve pain and function at long-term follow-up compared to a non-surgical comparison [25,31,32], with one study also reporting short-term benefits of surgery [25]. In contrast, two studies comparing surgical versus non-surgical interventions, provided evidence of no between-group difference at short- or long-term follow-up for pain or function [26,27]. However, the evidence is limited in both volume and quality and as such, no firm conclusions can be drawn regarding the efficacy or safety of any surgical approach.

## Overall completeness and applicability of evidence

We undertook a systematic search of multiple databases, for both published and unpublished studies, in addition to consulting experts in the field with a view to reducing the risk of omitting relevant evidence. However, completeness of the data may have been affected by a number of issues.

Following initial screening of the databases, we attempted to contact corresponding authors of three potential studies to obtain further information on their research. We were unable to source three registered studies despite email attempts to study authors: one of which had an available abstract but no full-text paper (comparing small needle scalpel under ultrasonography guidance with a 'traditional knife' approach in PF) [33] (n = 234); one registered clinical trial reported to be 'completed' but we were unable to locate the published trial (National Clinical Trial number: NCT00189592, n = 45;) and one registered clinical trial reported as 'ongoing' (Thai Clinical Trials Registry Number: 20180212001, planned sample size n = 40). In the absence of further contact from the trial authors, these studies are awaiting classification We contacted four authors of included studies for additional information. Of those, the authors of one study [28] did not respond. It is therefore possible that, despite these efforts, this review may be missing relevant data. A search of current trials registers did not reveal evidence that any current randomised control trials, investigating the effect of surgical interventions in people with PF, are being undertaken at present.

Across all studies, the methods reported relating to the measuring and reporting of adverse events were inadequate, inconsistent or absent. We found deficiencies in the presentation of data relating to adverse events in some studies, preventing reasonable conclusions from being made. Although the adverse events described were minor in nature, when considering the small size of all included studies, the safety of surgical interventions for PF must be viewed with caution. From the information available, the potential for more serious adverse events following surgery for PF cannot be excluded. To better appreciate the incidence and nature of adverse events, data from non-randomised studies and other sources such as surgical registers could be valuable, however, that was outside the scope of this review.

In the absence of any placebo controlled trials within this review it is not possible to determine whether effects observed were due to the intervention received, the placebo effect, or due to the natural progression of the condition with time. Furthermore, none of the studies in this review included data on cost-effectiveness, hence, we are not enable to make conclusions as to which interventions, surgical or non-surgical, are most cost-effective for people with PF.

## Certainty of evidence

We judged all studies to be at unclear or high RoB across multiple domains. All studies were judged as high RoB for performance bias, with six of the eight studies also reporting high RoB in at least two of the key domains of performance bias, detection bias, and reporting bias. These multiple sources of potential bias are likely to have contributed to any observed effects. This, in addition to the fact that each comparison consisted of one small study makes it appropriate to judge the evidence as of very low certainty.

None of the trials included within this review were placebo controlled. Placebo controlled trials demonstrate the efficacy of surgical interventions [34] and are of particular importance when there is poor evidence for the efficacy of a procedure [35], such as surgical interventions for people with PF. Placebo controlled trials would be both possible and important in this study population to ensure people with PF receive effective treatment.

While it would not have been possible to blind the surgeons conducting the procedures described within this review, for most studies, it would also not have been possible to blind participants to the intervention. However, participant and assessor blinding was potentially possible in two studies, but was not implemented [24,28], both deemed to be at 'unclear' RoB, which compared surgical techniques with similar incision sites. Although blinding of assessors who administer outcome measures is often possible, this was not always achieved in a number of included studies. Reducing these biases would likely improve the certainty of available evidence.

Of the eight studies presented within this review, all trials had small sample sizes, six with fewer than 50 participants. This presents a risk of small study bias, in which small published studies tend to show inflated estimates of effect [36,37]. Dechartres [36] demonstrated, in a review of meta-analyses, that such trials, with fewer than 50 participants, reported effect estimates 23% larger than effect estimates from studies with sample sizes greater than 50. Such sources of potential bias may explain any reported positive effects in the included studies.

We judged the overall quality of the evidence included within this review to be 'very low' according to GRADE criteria. Conclusions from individual studies should therefore be interpreted with very limited confidence.

## Potential biases in the review process

To identify all eligible trials we instigated a multiple database search strategy, without language restrictions. Following thorough searches, we are confident that this review represents the current evidence relating to the effect of surgical intervention in the management of people with PF. However, it may be possible that some key literature has been missed.

The study protocol inclusion criteria stipulated that 'any surgical procedure' would be included within the review. Following discussion with two Orthopaedic Foot and Ankle Consultants and co-authors (AR and TS) the medical procedures of cryosurgery, coblation and radiofrequency ablation, although minimally invasive, were not felt to qualify as surgical procedures due to the fact that they are usually conducted via the insertion of a very small probe or cannula through the skin and are unlikely to involve a full incision of a patients skin. Hence, we excluded such medical procedures from the review process; we cannot comment on the effectiveness of such procedures.

We noted limitations of the trials in evaluating safety. Although the adverse events described were minor, the potential for more serious adverse events following surgery cannot be excluded [38]. These issues are amplified by the predominance of small studies. There may be value in a review focused on adverse events that includes observational studies for adverse events [38], and also in developing registry data.

### Agreements and disagreements with other studies or reviews

To our knowledge this is the first systematic review to examine the effectiveness of all surgical procedures in the management of PF. Mao et al [39] conducted a systematic review and meta-analysis of EPF for PF. They included randomised and non-randomised studies, and case series whereas the current systematic review included evidence from RCTs. Of the 12 studies included within Mao et al.'s review [39], only one was an RCT [27]–this paper was also included in the current review. The majority of studies included within Mao's review were case series. The authors similarly concluded that there is insufficient high-level evidence to enable clinical recommendations to be made.

Malahias et al. published a self-described 'current concept' review of clinical outcomes following EPF to determine the safety and effectiveness of this procedure in people with PF [40]. Following a search of the databases they reviewed 15 studies. Only 2 included studies were RCTs [26,27], both of which were included in the current review. Malahias et al. [40] concluded there was weak evidence that EPF was safe and effective in people with PF, however, note that from the available evidence EPF cannot be considered superior to other management options that are less invasive. Interestingly, in the review by Malahias et al, including a greater number of studies of all levels of evidence demonstrated additional, and likely more functionally limiting adverse events, including calcaneal stress fracture, and chronic longitudinal arch strain following EPF.

## Authors conclusions

### Implications for practice

Surgical interventions are offered to people with persistent, intractable PF, to help reduce pain and improve function. However, this review has highlighted a paucity of high-quality evidence regarding their effectiveness. Due to the scarcity of evidence reporting on adverse events, it is also not possible to make confident conclusions regarding the safety of PF surgical interventions. People with PF, should be made aware of these gaps in knowledge within the current evidence when considering surgery as a management option.

On the basis of such evidence, it is not possible to make any clinical recommendations. This review has demonstrated that there is no high certainty evidence to support or refute the use of surgical interventions in the management of PF.

### Implications for research

The current evidence base investigating the effectiveness of surgery in people with PF consists of small studies with high RoB. This systematic review has demonstrated the need for larger high-quality studies to be conducted in this clinical population. Surgery has a large potential for creating placebo effects, hence, a placebo group within surgical comparison trials would be both possible and essential. Such future research would benefit from a multicentre approach to ensure recruitment from larger populations. Pre-registration of the study protocol is essential and future trials should meet contemporary standards of best practice with regards reporting. Outcomes must be patient-centred, with a focus on valid and reliable outcomes for function and pain. The methodology for capturing and reporting adverse events must be clear, consistent and complete. Previous research has demonstrated ongoing heel pain greater than one year post onset of PF symptoms, hence, due to the natural history of the condition, follow-up of greater than one year would be recommended. Finally, outcome reporting must be to a higher, more complete standard, including effect sizes and confidence intervals for all outcomes at each time-point. Until research fulfilling these criteria is completed, the effectiveness of surgery in the management of PF will remain unclear.

## Supporting information

**S1 Checklist. PRISMA 2020 checklist.**
(DOCX)

**S1 Appendix. Medline search strategy.**
(DOCX)

**S2 Appendix. Assessment of risk of bias in included studies.**
(DOCX)

**S3 Appendix. Certainty of the evidence.**
(DOCX)

**S4 Appendix. Characteristics of excluded trials and trials awaiting classification.**
(DOCX)

**S5 Appendix. Characteristics of the included studies.**
(DOCX)

**S6 Appendix. Risk of bias in the included studies.**
(DOCX)

**S7 Appendix. Effects of interventions: Secondary outcome measures.**
(DOCX)

## Author Contributions

**Conceptualization:** C. Sian MacRae, Neil E. O'Connell.

**Data curation:** C. Sian MacRae, Neil E. O'Connell.

**Formal analysis:** C. Sian MacRae, Neil E. O'Connell.

**Methodology:** C. Sian MacRae, Andrew J. Roche, Tim J. Sinnett, Neil E. O'Connell.

**Writing – original draft:** C. Sian MacRae, Andrew J. Roche, Tim J. Sinnett, Neil E. O'Connell.

**Writing – review & editing:** C. Sian MacRae, Andrew J. Roche, Tim J. Sinnett, Neil E. O'Connell.

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
