## [Decision Letter · Decision Letter 0]

14 Feb 2022

PONE-D-21-22044What is the evidence for efficacy, effectiveness and safety of surgical interventions for plantar fasciopathy?  A systematic review.PLOS ONE

Dear Dr. Macrae,

Thank you for submitting your manuscript to PLOS ONE. After careful consideration, we feel that it has merit but does not fully meet PLOS ONE’s publication criteria as it currently stands. Therefore, we invite you to submit a revised version of the manuscript that addresses the points raised during the review process.

We look forward to receiving your revised manuscript.

Kind regards,

Osama Farouk

Academic Editor

PLOS ONE

Journal Requirements:

2. Please update search and results to include relevant studies published since June 2020.

5. Please upload a new copy of Figures 2 and 3 as the detail is not clear. Please follow the link for more information: " ext-link-type="uri" xlink:type="simple">https://blogs.plos.org/plos/2019/06/looking-good-tips-for-creating-your-plos-figures-graphics/"
" ext-link-type="uri" xlink:type="simple">https://blogs.plos.org/plos/2019/06/looking-good-tips-for-creating-your-plos-figures-graphics/"

Reviewers' comments:

Reviewer's Responses to Questions

**Comments to the Author**

1. Is the manuscript technically sound, and do the data support the conclusions?

Reviewer #1: Yes

Reviewer #2: Yes

2. Has the statistical analysis been performed appropriately and rigorously? 

Reviewer #1: I Don't Know

Reviewer #2: I Don't Know

3. Have the authors made all data underlying the findings in their manuscript fully available?

Reviewer #1: Yes

Reviewer #2: Yes

4. Is the manuscript presented in an intelligible fashion and written in standard English?

Reviewer #1: Yes

Reviewer #2: Yes

5. Review Comments to the Author

Reviewer #1: The work is devoted to an interesting topic. Plantar fasciopathy is one of the most common problems in in everyday orthopedic practice.

The methodology of the study seems to me to be proper.

The results show that we still do not know much about surgical therapy.

Reviewer #2: Dear Authors,

thank you for this intersting manuscript about a very common pathology with no clear defined treatment algorithms.

The manuscript is written in good english, the formatting is correct.

I do have only a minor remark:

You included the study by Sadak about the lateral plantar nerve release.

I do not understand why you included this study, as it differs largely from the other studies. In my opinion, it also soughts to treat a different pathology. Lateral plantar nerve release (Baxters nerve) is close to the plantar fascia and its mainly degenerative insertional tendinitis, but represents a different pathology with nerve entrapment. No doubt, both pathologies can can be present at the same time, but in the end, they are somewhat different and insertional tendinitis of the plantar fasica can not be treted by release of the lateral plantar nerve.

Please respond.

6. PLOS authors have the option to publish the peer review history of their article (what does this mean?). If published, this will include your full peer review and any attached files.

Reviewer #1: No

Reviewer #2: No

---

## [Author Response · Author response to Decision Letter 0]

17 Mar 2022

Dear Academic Editor and Reviewers,

Thank you for your time in reviewing our systematic review entitled “What is the evidence for efficacy, effectiveness and safety of surgical interventions for plantar fasciopathy? A systematic review” and for the helpful feedback provided.

We have addressed each point raised and a summary of our responses can be found below:

Point 1: we have ensured that the manuscript meets the PLOS ONE style requirements. Any changes made can be seen within the uploaded file labelled 'Revised Manuscript with Track Changes'.

Point 2: we have updated our search up to February 25th 2022 and made relevant changes within the manuscript and Figure 1 accordingly. Any changes made can be seen within the uploaded file labelled 'Revised Manuscript with Track Changes'. In summary we found no additional published trials, but the search did reveal a recently registered trial protocol.

Point 3: We have added further detail to the ‘Data Availability Statement’ further clarifying the exact location of the minimal data set underlying the results described, specifically that the minimal data set can be found within the document in Tables 1 and 2 in addition to that in the Supporting Information Files. If there is any further data that you would wish us to include please let us know.

“Data Availability Statement: The minimal data set underlying the results described can be found within the manuscript in Table 1 and 2, and within the Supporting Information Files.” 

Point 4: The corresponding author has been altered to Neil O’Connell.

Point 5: New copies of Figures 2 and 3 have been uploaded.

Point 6: Captions for our Supporting Information Files have been added at the end of the manuscript, and in-text citations have been updated to match accordingly.

Point 7: The references list has been reviewed and in some cases references have been adapted to ensure they meet the PLOS ONE style requirements eg additional author names and DOI’s added. Any modifications made can be seen within the uploaded file labelled 'Revised Manuscript with Track Changes'. We did not find any references that had been retracted within the reference list other than an update to the NICE Plantar Fasciitis Clinical Knowledge Summary which we updated from 2015 to 2020.

Reviewers Comments:

Point 5:

In response to Reviewer #2, thank you for your comment regarding our reasons for including the Sadak study within our systematic review. The Sadak study was included in our review as their inclusion criteria (people with “resistant plantar fasciitis after ≥6 months of conservative management. The participants in their study had their diagnosis confirmed by the presence of a calcaneal spur and perifascial oedema, and increased plantar fascial thickness of 4 mm”) met our inclusion criteria (“studies involving adults, aged 18 years or older, diagnosed with plantar fascioapthy, or with an alternative diagnostic label for this condition e.g. plantar fasciitis, plantar heel pain. We included studies regardless of symptom duration or whether radiological diagnostic imaging had been employed”). Of note and of interest, even though they appear to have chosen a technique that may seem more indicated for people presenting with neural symptoms, such as Baxters neuropathy, the authors do not describe neural symptoms as an additional inclusion criteria for their study.

We thank you for your time in considering our revisions and comments and look forward to hearing from you.

Kind regards,

Sian

Sian MacRae PhD

Advanced Physiotherapy Practitioner

---

## [Decision Letter · Decision Letter 1]

4 May 2022

What is the evidence for efficacy, effectiveness and safety of surgical interventions for plantar fasciopathy?  A systematic review.

PONE-D-21-22044R1

Dear Dr. O'Connell,

We’re pleased to inform you that your manuscript has been judged scientifically suitable for publication and will be formally accepted for publication once it meets all outstanding technical requirements.

Kind regards,

Osama Farouk

Academic Editor

PLOS ONE

Additional Editor Comments (optional):

Reviewers' comments:

Reviewer's Responses to Questions

**Comments to the Author**

1. If the authors have adequately addressed your comments raised in a previous round of review and you feel that this manuscript is now acceptable for publication, you may indicate that here to bypass the “Comments to the Author” section, enter your conflict of interest statement in the “Confidential to Editor” section, and submit your "Accept" recommendation.

Reviewer #1: All comments have been addressed

2. Is the manuscript technically sound, and do the data support the conclusions?

Reviewer #1: Yes

3. Has the statistical analysis been performed appropriately and rigorously? 

Reviewer #1: I Don't Know

4. Have the authors made all data underlying the findings in their manuscript fully available?

Reviewer #1: Yes

5. Is the manuscript presented in an intelligible fashion and written in standard English?

Reviewer #1: Yes

6. Review Comments to the Author

Reviewer #1: The authors have adequately addressed the prior review. The topic is of interest for daily practise.

7. PLOS authors have the option to publish the peer review history of their article (what does this mean?). If published, this will include your full peer review and any attached files.

Reviewer #1: No

---

## [Editor Report · Acceptance letter]

10 May 2022

PONE-D-21-22044R1 

What is the evidence for efficacy, effectiveness and safety of surgical interventions for plantar fasciopathy?  A systematic review. 

Dear Dr. O'Connell:

I'm pleased to inform you that your manuscript has been deemed suitable for publication in PLOS ONE. Congratulations! Your manuscript is now with our production department. 

Kind regards, 

on behalf of

Dr. Osama Farouk 

Academic Editor

PLOS ONE